# A Quasistatic Derivation of Optimization Algorithms' Exploration on the Minima Manifold

## Abstract

A quasistatic approach is proposed to derive the optimization algorithms' effective dynamics on the manifold of minima when the iterator oscillates around the manifold. Compared with existing strict analysis, our derivation method is simple and intuitive, has wide applicability, and produces easy-to-interpret results. As examples, we derive the manifold dynamics for SGD, SGD with momentum (SGDm) and Adam with different noise covariances, and justify the closeness of the derived manifold dynamics with the true dynamics through numerical experiments. We then use minima manifold dynamics to study and compare the properties of optimization algorithms. For SGDm, we show that scaling up learning rate and batch size simultaneously accelerates exploration without affecting generalization, which confirms a benefit of large batch training. For Adam, we show that the speed of its manifold dynamics changes with the direction of the manifold, because Adam is not rotationally invariant. This may cause slow exploration in high dimensional parameter spaces.

## 1 Introduction

The ability of stochastic optimization algorithms to explore among (global) minima is believed to be one of the essential mechanisms behind the good generalization performance of stochastically trained over-parameterized neural networks. Until recently, research on this topic has focused on how the iterator jumps between the attraction basins of many isolated minima and settles down around the flattest one Xie et al. (2020); Nguyen et al. (2019); Dai & Zhu (2020); Mori et al. (2021). However, for over-parameterized models, the picture of isolated minima is not accurate, since global minima usually form manifolds of connected minima Cooper (2018). In addition to crossing barriers and jumping out of the attraction basin of one minima, the optimizer also moves along *minima manifold* and search for better solutions Wang et al. (2021). Hence, understanding how optimization algorithms explore along the minima manifold is crucial to understanding how stochastic optimization algorithms are able to find generalizing solutions for over-parameterized neural networks.

Some recent works have begun to examine the exploration dynamics of Stochastic Gradient Descent (SGD) along minima manifolds. Many of these works have identified how a change of flatness in the minima manifold adds a driving force to SGD as it oscillates around the minima. For example, Damian et al. (2021) considered an SGD training a neural network with label noise, and showed that the optimizer can find the flattest minimum among all global minima. A more recent work Li et al. (2021b) derived an effective stochastic dynamics for SGD on the manifold. The results in Li et al. (2021b) show that (when the learning rate tends to zero) the changing flatness can give a force to the SGD iterator along the minima manifold and induce a slow dynamics on the manifold that helps the SGD move to the vicinity of flatter minima.

In this work, we study the same questions of the flatness-driven exploration along the minima manifold. Instead of searching for a strict proof, we focus on simple and intuitive ways to derive the *manifold dynamics*. Specifically, we propose a *quasistatic approach* to derive the manifold dynamics for different optimization algorithms and stochastic noises. The main technique of our derivation is a time-scale decomposition of the motions perpendicular to and parallel with the minima manifold, which we call the *normal component* and the *tangent component*, respectively. We treat the normal component as infinitely faster than the tangent component, and thus it is always at equilibrium given

the tangent component. The effective dynamics of the tangent component, i.e. the manifold dynamics, is obtained by taking the expectation over the equilibrium distribution of the normal component. The main step in our analysis involves deriving the equilibrium covariance of an SDE. Compared with the theoretical analysis in Li et al. (2021b), our derivation and results are simpler and easier to interpret, and clearly identifies the roles played by each component of the optimization algorithm (noise covariance, learning rate, momentum). The following simple example demonstrates the main idea of the our derivations.

**A simple illustrative example:** Consider a loss function $f(x, y) = h(x)y^2$, where $h(x) > 0$ is a differentiable function of $x$. The global minima of this function lie on the $x$-axis, forming a flat manifold, and $h(x)$ controls the flatness of the loss function at any minimum $(x, 0)$. Let $\boldsymbol{z} = [x, y]^T$. We consider an SGD approximated by SDE Li et al. (2017)Li et al. (2021a)

$$d\boldsymbol{z}_t = -\nabla f(\boldsymbol{z}_t)dt + \sqrt{\eta}D(\boldsymbol{z}_t)d\boldsymbol{W}_t, \tag{1}$$

where $\eta$ is the learning rate, $D$ is the square root of the covariance matrix of the gradient noise, and $\boldsymbol{W}_t$ is a Brownian motion. For the convenience of presentation, for points $(x, y)$ that are close to the x-axis, we assume the noise covariance aligns with the Hessian of the loss function at $(x, 0)$, i.e. $D^2(\boldsymbol{z}) = \frac{\sigma^2}{2}Hf(x, 0) = \begin{bmatrix} 0 & 0 \\ 0 & \sigma^2 h(x) \end{bmatrix}$, where $\sigma > 0$ is a scalar. Then, the SDE equation 1 can be written as

$$dx_t = -h'(x_t)y_t^2 dt, \quad dy_t = 2h(x_t)y_t dt + \sigma\sqrt{\eta h(x_t)}dW_t,$$

with $W_t$ being a 1-D Brownian motion. When $y_t$ is close to 0, the speed of $x_t$ is much slower than $y_t$ because of the $y_t^2$ in the dynamics of $x$ is much smaller than the $y_t$ in the dynamics of $y$. When this separation of speed is large, the dynamics above can be approximated by the following quasistatic dynamics

$$dx_t = -\lim_{\tau \to \infty} \mathbb{E}_{y_\tau} h'(x_t)y_\tau^2 dt, \quad dy_\tau = 2h(x_t)y_\tau d\tau + \sqrt{\eta h(x_t)}\sigma dW_\tau. \tag{2}$$

which assumes $y$ is always at equilibrium given $x_t$. Solving the Ornstein–Uhlenbeck process 2, we know the equilibrium distribution of $y_\tau$ is $\sim N(0, \frac{\eta\sigma^2}{4})$, and hence the manifold dynamics is

$$\frac{dx_t}{dt} = -\frac{\eta\sigma^2 h'(x_t)}{4}.$$

This derivation shows the slow effective dynamics along the manifold is a gradient flow minimizing the flatness $h(x)$. This simple quasistatic derivation reveals the flatness-driven motion of SGD along the minima manifold, and recovers the same dynamics as given by Li et al. (2021b) in this specific case. On the left panel of Figure 1, we show an SGD trajectory for $f(x, y) = (1 + x^2)y^2$, illustrating the exploration along the manifold due to the oscillation in the normal space. On the right panel we verify the closeness of the manifold dynamics with the true SGD trajectories for the same objective function. The "Hessian noise" and "Isotropic noise" represent noises whose covariance are the Hessian matrix of $f$ (as analyzed above) and the identity matrix (covered by the analysis in Section 2), respectively.

**Theoretical applications of the manifold dynamics:** The minima manifold dynamics of optimization algorithms can be used as a tool to study and compare the behaviors of optimization algorithms. In Section 3, we illustrate how our derivations can be applied to study the behavior of SGD on a matrix factorization problem. Two more interesting applications are discussed Section 4 and 5. In Section 4, we focus on SGDm, and study the role played by the learning rate, batch size, and momentum coefficient in its manifold dynamics. Based on the analysis, we explore approaches to reliably accelerate the manifold dynamics, which may help accelerate training. Especially, we show that scaling up learning

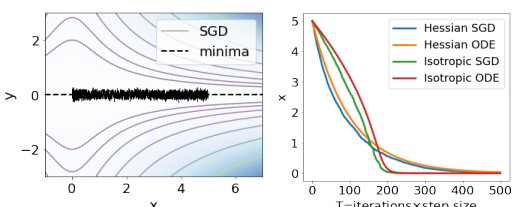

Figure 1: **(left)** The trajectory of SGD (real dynamics) with Hessian noise initialized from $(5, 0)$. **(middle)** The x-coordinate of the real dynamics and the manifold dynamics for SGD with Hessian and isotropic noises.

rate and batch size simultaneously accelerates exploration without affecting generalization, which confirms a benefit of large batch training. In Section 5, we study adaptive gradient methods, and show that the speed of the manifold dynamics of Adam Kingma & Ba (2014) changes with the direction of the manifold, because Adam is not rotationally invariant. When the manifold does not align well with some axis direction, the exploration of Adam along the manifold is even slower than SGD with the same learning rate. This shows the sensitivity of Adam (and other adaptive gradient methods) to the parameterization and a potential weakness of Adam on the exploration among global minima.

**Additional related work:**   Besides the important related works discussed above, our work is related broadly with the topic of implicit regularization of optimization algorithms for overparameterized models. Such works include global minima selection Wu et al. (2018); Ma & Ying (2021), the max-margin bias of some homogeneous models Soudry et al. (2018); Lyu & Li (2019); Lyu et al. (2021), and others such as Gunasekar et al. (2017); Li et al. (2018); Woodworth et al. (2020); Neyshabur et al. (2017) (not to be comprehensive). One work that is worth mentioning is Blanc et al. (2020), in which the implicit regularization effect of SGD under label noise is studied by understanding the SGD as a Ornstein-Uhlenbeck process (OU process), which also plays an important role in our derivation.

For applications, we give theoretical explanations for the benefit of large batch training, which is an important and extensively studied empirical topic You et al. (2019; 2017); Hoffer et al. (2017); Geiping et al. (2021). Our study on adaptive gradient methods concerns the family of widely used methods which adjusts the learning rate adaptively (and usually coordinatewise) Duchi et al. (2011); Tieleman & Hinton (2012); Kingma & Ba (2014).

## 2   THE QUASISTATIC DERIVATION FOR MANIFOLD DYNAMICS

In this section, we introduce our quasistatic approach for deriving minima manifold dynamics, which is an effective exploration dynamics of optimization algorithms on minima manifolds.

**Notations:**   Let $\mathcal{M}$ be a smooth manifold in $\mathbb{R}^d$ with Euclidean metric. Throughout this paper, we use $\boldsymbol{z}$ to denote points in $\mathbb{R}^d$, including $\mathcal{M}$, and use $\mathbf{x}$ and $\boldsymbol{y}$ to denote components of $\boldsymbol{z}$ used in the quasistatic derivation for speed separation. For any $\boldsymbol{z} \in \mathcal{M}$, let $T_{\boldsymbol{z}}\mathcal{M}$ be the tangent space of $\mathcal{M}$ at $\boldsymbol{z}$. Let $T_{\boldsymbol{z}}\mathcal{M}^{\perp}$ be the orthogonal complement of $T_{\boldsymbol{z}}\mathcal{M}$, which is the normal space of $\mathcal{M}$ at $\boldsymbol{z}$. Let $\mathcal{P}_{\mathcal{M}}$ be the projection operator onto $\mathcal{M}$, i.e. for any $\boldsymbol{z} \in \mathbb{R}^d$, $\mathcal{P}_{\mathcal{M}}\boldsymbol{z}$ gives the closest point on $\mathcal{M}$ to $\boldsymbol{z}$ (if exists and unique). $\mathcal{P}_{\mathcal{M}}$ is well defined when $\boldsymbol{z}$ is close to $\mathcal{M}$ Lee (2013). In the paper, we ignore the subscript $\mathcal{M}$ for $\mathcal{P}_{\mathcal{M}}$ when there is no confusion.

**Problem settings**   We consider a boundless $k$-dimensional smooth manifold $\mathcal{M}$ in $\mathbb{R}^d$, formed by (local or global) minima of a function $f : \mathbb{R}^d \to \mathbb{R}$. Since we are interested in the behavior of stochastic optimizers near the minima manifold $\mathcal{M}$, we ignore the landscape of $f$ when $\mathbf{x}$ is far away from $\mathcal{M}$, and only consider a quadratic expansion of $f$ at $\mathcal{M}$. Specifically, let $H(\cdot) : \mathcal{M} \to \mathbb{R}^{d \times d}$ be a function on $\mathcal{M}$ that gives the Hessian of the loss function on the minima manifold. For any $\boldsymbol{z} \in \mathcal{M}$, we assume $H(\boldsymbol{z})$ is positive semidefinite, and its 0-eigenspace is exactly $T_{\boldsymbol{z}}\mathcal{M}$. The loss functions that we consider take the form

$$f(\boldsymbol{z}) = (\boldsymbol{z} - \mathcal{P}\boldsymbol{z})^T H(\mathcal{P}\boldsymbol{z})(\boldsymbol{z} - \mathcal{P}\boldsymbol{z}), \ \ \boldsymbol{z} \in \mathbb{R}^d. \tag{3}$$

For the optimization dynamics, we start from SGD, approximated by the following SDE Li et al. (2017)Li et al. (2021a)

$$d\boldsymbol{z}_t = -\nabla f(\boldsymbol{z}_t)dt + \sqrt{\eta}D(\mathcal{P}\boldsymbol{z}_t)dW_t, \tag{4}$$

where $W_t$ is a Brownian motion in $\mathbb{R}^d$ and $D : \mathcal{M} \to \mathbb{R}^{d \times d}$ is the square root of the noise covariance. Later we will extend our analysis to SGDm and Adam. Strictly speaking, the noise of SGD depends on $\boldsymbol{z}$, which is not always on $\mathcal{M}$. However, in settings that we study, we assume it only depends on $\mathcal{P}\boldsymbol{z}$ because $\boldsymbol{z}$ is close to $\mathcal{M}$.

### 2.1   FLAT MANIFOLD

We start from the case in which the minima manifold $\mathcal{M}$ is flat. The derivation in this case is similar to the 2-D example given in the introduction. For any $\boldsymbol{z} \in \mathbb{R}^d$, let $\boldsymbol{z} = [\mathbf{x}^T, \boldsymbol{y}^T]^T$, with $\mathbf{x} \in \mathbb{R}^k$ and $\boldsymbol{y} \in \mathbb{R}^{d-k}$. Without loss of generality, we assume $\mathcal{M} = \{\boldsymbol{z} = [\mathbf{x}^T, \boldsymbol{y}^T]^T : \boldsymbol{y} = 0\}$,

i.e. $\mathcal{M}$ is the linear subspace formed by the first $k$ axes. Then, for any $\boldsymbol{z} = \begin{bmatrix} \mathbf{x} \\ \boldsymbol{y} \end{bmatrix}$, we have

$\mathcal{P}\boldsymbol{z} = \begin{bmatrix} \mathbf{x} \\ \mathbf{0} \end{bmatrix}$ and $\boldsymbol{z} - \mathcal{P}\boldsymbol{z} = \begin{bmatrix} \mathbf{0} \\ \boldsymbol{y} \end{bmatrix}$. For the loss function $f$, since we assume its Hessian has zero

eigenvalues along the tangent space of $\mathcal{M}$, the Hessian must take the form $H(\mathbf{x}) = \begin{bmatrix} 0 & 0 \\ 0 & \tilde{H}(\mathbf{x}) \end{bmatrix}$

with $\tilde{H}(\mathbf{x}) \in \mathbb{R}^{(d-k)\times(d-k)}$. Here, we can treat $H$ as a function of $\mathbf{x}$ because it is defined on $\mathcal{M}$. Hence, written as a function of $\mathbf{x}$ and $\boldsymbol{y}$, the loss function 3 becomes

$$f(\mathbf{x}, \boldsymbol{y}) = \boldsymbol{y}^T \tilde{H}(\mathbf{x})\boldsymbol{y}. \tag{5}$$

Next, we rewrite the SDE 4 using $\mathbf{x}$ and $\boldsymbol{y}$. Again, since the noise coefficient $D$ in 4 is defined on $\mathcal{M}$, it can be treated as a function of $\mathbf{x}$. For any $\mathbf{x}$, let $D(\mathbf{x}) = \begin{bmatrix} \tilde{D}_{11}(\mathbf{x}) & \tilde{D}_{12}(\mathbf{x}) \\ \tilde{D}_{21}(\mathbf{x}) & \tilde{D}_{22}(\mathbf{x}) \end{bmatrix} \in \mathbb{R}^{d\times d}$ with

$\tilde{D}_{11}(\mathbf{x}) \in \mathbb{R}^{k\times k}$. Then, $\tilde{D}_{11}$ represents the noise in the tangent space, $\tilde{D}_{22}$ represents the noise in the normal space, and $\tilde{D}_{12}, \tilde{D}_{21}$ represent the interaction of the tangent space and normal space noises. In many cases, the noise covariance matrix of SGD aligns with the Hessian Mori et al. (2021). Hence, $\tilde{D}_{22}$ dominates the other three components. (Actually, when the noise covariance strictly aligns with the Hessian, only $\tilde{D}_{22}$ is nonzero.) In the following, we assume the interactions $\tilde{D}_{12}$ and $\tilde{D}_{21}$ are 0, while $\tilde{D}_{11}$ can still be nonzero. Then, by the form of the loss function in 5, the SDE 4 can be written as the following system of $\mathbf{x}$ and $\boldsymbol{y}$:

$$dx_t = -\boldsymbol{y}_t^T \partial_{\mathbf{x}} \tilde{H}(\mathbf{x}_t)\boldsymbol{y}_t dt + \sqrt{\eta}\tilde{D}_{11}(\mathbf{x}_t)dW_t^{(1)}$$
$$d\boldsymbol{y}_t = -2\tilde{H}(\mathbf{x}_t)\boldsymbol{y}_t dt + \sqrt{\eta}\tilde{D}_{22}(\mathbf{x}_t)dW_t^{(2)}, \tag{6}$$

where $\partial_{\mathbf{x}}\tilde{H}(\mathbf{x}_t)$ is a $(d-k)\times(d-k)\times k$ tensor containing all partial derivatives of $\tilde{\mathbf{x}}$. When $\boldsymbol{z}$ is close to $\mathcal{M}$, $\boldsymbol{y}$ is small, in which case the dynamics of $\boldsymbol{y}$ is much faster than that of $\mathbf{x}$, because the drift term for $\mathbf{x}$ depends quadratically with $\boldsymbol{y}$ while the drift term for $\boldsymbol{y}$ only depends linearly with $\boldsymbol{y}$. Therefore, we can use a quasistatic dynamics to approximate the original dynamics. The quasistatic dynamics assumes that $\boldsymbol{y}$ is always at equilibrium:

$$dx_t = -\lim_{\tau\to\infty} \boldsymbol{y}_\tau^T \partial_{\mathbf{x}}\tilde{H}(\mathbf{x}_t)\boldsymbol{y}_\tau dt + \sqrt{\eta}\tilde{D}_{11}(\mathbf{x}_t)dW_t^{(1)} \tag{7}$$

$$d\boldsymbol{y}_\tau = -2\tilde{H}(\mathbf{x}_t)\boldsymbol{y}_t dt + \sqrt{\eta}\tilde{D}_{22}(\mathbf{x}_t)dW_\tau^{(2)}. \tag{8}$$

Fixing $\mathbf{x}_t$, the dynamics for $\boldsymbol{y}_\tau$ is a linear SDE. We have $\boldsymbol{y}_\tau$ as $\lim_{\tau\to\infty}\mathbb{E}\boldsymbol{y}_\tau = 0$ and $\lim_{\tau\to\infty}\mathbb{E}\boldsymbol{y}_\tau\boldsymbol{y}_\tau^T = V_t$, where $V_t \in \mathbb{R}^{(d-k)\times(d-k)}$ satisfies $\tilde{H}(\mathbf{x}_t)V_t + V_t\tilde{H}(\mathbf{x}_t) = \eta\tilde{D}_{22}(\mathbf{x}_t)\tilde{D}_{22}(\mathbf{x}_t)^T/2$. The derivations here are standard. Readers can refer to textbooks or lecture notes such as Herzog (2013). Substituting the moments into the $\boldsymbol{u}$-dynamics in 7, we have

$$dx_t = -\sum_{i,j=1}^{n-k} (V_t)_{ij}\nabla_{\mathbf{x}}(\tilde{H}(\mathbf{x}_t)_{ij})dt + \sqrt{\eta}\tilde{D}_{11}(\mathbf{x}_t)dW_t^{(1)}. \tag{9}$$

Understanding $\mathbf{x}$ as a vector on $\mathcal{M}$, equation 9 gives the effective manifold dynamics on $\mathcal{M}$. This result recovers the simple example in the introduction if we take $f(x,y) = h(x)y^2$ and $D^2 = \sigma^2 H/2$.

## 2.2 GENERAL MANIFOLD

For general smooth manifold, the manifold dynamics can be derived locally by approximating $\mathcal{M}$ using a flat manifold. The resulting dynamics is different from 9 only in that the gradients are taken on the manifold. To see this, consider any point $\boldsymbol{z}_0 \in \mathcal{M}$. Without loss of generality, we assume there exists $\mathbf{x}_0 \in \mathbb{R}^k$, such that $\boldsymbol{z}_0 = \begin{bmatrix} \mathbf{x}_0 \\ \mathbf{0} \end{bmatrix}$, and $T_{\boldsymbol{z}_0}\mathcal{M} = \left\{ \begin{bmatrix} \mathbf{x} \\ \mathbf{0} \end{bmatrix} : \mathbf{x} \in \mathbb{R}^k \right\}$. Because $\mathcal{M}$ is a smooth manifold, around $\boldsymbol{z}_0$ the projection operator onto $T_{\boldsymbol{z}_0}\mathcal{M}$, denoted by $P_{\boldsymbol{z}_0}$, induces an 1-1 map between $\mathcal{M}$ and $T_{\boldsymbol{z}_0}\mathcal{M}$, and we have $\|\boldsymbol{z} - P_{\boldsymbol{z}_0}\boldsymbol{z}\| = \mathcal{O}(\|\boldsymbol{z} - \boldsymbol{z}_0\|^2)$. Let $P_{\boldsymbol{z}_0}^{-1} : T_{\boldsymbol{z}_0}\mathcal{M} \to \mathcal{M}$ be the inverse of this 1-1 map. With an abuse of notations, for $\boldsymbol{z} = [\mathbf{x}^T, \mathbf{0}]^T \in T_{\boldsymbol{z}_0}\mathcal{M}$, we sometimes also use $P_{\boldsymbol{z}_0}^{-1}\mathbf{x}$ to denote $P_{\boldsymbol{z}_0}^{-1}\boldsymbol{z}$. Let $O_{\boldsymbol{z}_0} = \{\boldsymbol{e}_1, ..., \boldsymbol{e}_d\}$ be the standard orthonormal basis for $\mathbb{R}^d$,

with $\{e_1, ..., e_k\}$ being an orthonormal basis for $T_{z_0}\mathcal{M}$. Also because $\mathcal{M}$ is smooth, for any $z \in \mathcal{M}$ close to $z_0$, there exists an orthonormal basis $O_z = \{e_1^z, ..., e_d^z\}$ for $\mathbb{R}^d$ which is close to $O_{z_0}$, such that $\{e_1^z, ..., e_k^z\}$ form an orthonormal basis for $T_z\mathcal{M}$. Specifically, for any $1 \le i \le d$ we have $\|e_i^z - e_i\| = \mathcal{O}(\|z - z_0\|^2)$.

Now, consider SDE 4 with the loss function $f$ defined in 3. For any $z \in \mathcal{M}$ close to $z_0$, let $H(z) = \begin{bmatrix} 0 & 0 \\ 0 & \tilde{H}(z) \end{bmatrix}$ and $D(z) = \begin{bmatrix} \tilde{D}_{11}(z) & 0 \\ 0 & \tilde{D}_{22}(z) \end{bmatrix}$ be the Hessian and the noise coefficient matrix expressed in $O_z$. The zeros in the expressions are due to assumptions on $H$ and the noise, i.e. $T_z\mathcal{M}$ is in the 0-eigenspace of $H$, and there is no interaction between the tangent space noise and the normal space noise. Now, we define a companion loss function $\tilde{f}$ whose minima manifold is $T_{z_0}\mathcal{M}$ by $\tilde{f}(z) = y^T \tilde{H}(P_{z_0}^{-1}\mathbf{x})y$, where $z = [\mathbf{x}^T, y^T]^T$, and consider a companion SDE

$$d z_t = -\nabla \tilde{f}(z_t)dt + \sqrt{\eta}D(P_{z_0}^{-1}\mathcal{P}_{z_0}z_t)dW_t. \tag{10}$$

The SDE above approximates an SGD minimizing $\tilde{f}$, whose minima manifold is flat. Hence, using the results from the previous subsection, we can derive a manifold dynamics on $T_{z_0}\mathcal{M}$,

$$d\mathbf{x} = -\sum_{i,j=1}^{n-k} (V_t)_{ij} \nabla_{\mathbf{x}}(\tilde{H}(P_{z_0}^{-1}\mathbf{x})_{ij})dt + \sqrt{\eta}\tilde{D}_{11}(P_{z_0}^{-1}\mathbf{x})dW_t^{(1)}, \tag{11}$$

where $V_t$ is obtained by $\tilde{H}(P_{z_0}^{-1}\mathbf{x})V_t + V_t\tilde{H}(P_{z_0}^{-1}\mathbf{x}) = \eta\tilde{D}_{22}(P_{z_0}^{-1}\mathbf{x})\tilde{D}_{22}(P_{z_0}^{-1}\mathbf{x})^T/2$[1]. By the discussions above, around $z_0$ the SDE 4 is close to 10, because $\tilde{f}(z) \approx f(z)$ and $D(P_{z_0}^{-1}\mathcal{P}_{z_0}z_t) \approx D(\mathcal{P}z)$. Hence, the effective manifold dynamics of 4 is close to 11 in a neighborhood of $z_0$, and this approximation is better in smaller neighborhood of $z_0$. Therefore, at $z = z_0$ the manifold dynamics is approximately 11 taking $\mathbf{x} = \mathbf{x}_0$, which leads to

$$dz = -\sum_{i,j=1}^{n-k} V_{ij} \nabla_{\mathbf{x}}(\tilde{H}(z)_{ij})dt + \sqrt{\eta}\tilde{D}_{11}(z)dW_t^{(1)}.$$

Since $\nabla_{\mathbf{x}}(\tilde{H}(z)_{ij})$ are gradients in the tangent space $T_{z_0}\mathcal{M}$, under Euclidean metric they are gradients on the manifold at $z_0$. Hence, at $z_0$ the manifold dynamics can be written as

$$dz = -\sum_{i,j=1}^{n-k} V_{ij} \nabla_{\mathcal{M}}(\tilde{H}(z)_{ij})dt + \sqrt{\eta}\tilde{D}_{11}(z)dW_t^{(1)}. \tag{12}$$

Since the above analysis holds for any $z_0 \in \mathcal{M}$, the manifold dynamics of 4 is given by 12 at any $z$.

**Examples:** One interesting case is when the noise covariance matrix is proportional with the Hessian. In this case, $D^2(z) = \sigma^2 H(z)$ for any $z \in \mathcal{M}$. Since $\tilde{H}(z)$ contains all the nonzero eigenvalues of $H(z)$ and the nonzero eigenspace corresponds to $T_z\mathcal{M}^\perp$, we have $\tilde{D}_{11} = 0$ and $\tilde{D}_{22}(z) = \sigma\sqrt{\tilde{H}(z)}$. In this case, $V_t$ satisfies $\tilde{H}(z_t)V_t + V_t\tilde{H}(z_t) = \eta\sigma^2\tilde{H}(z_t)/2$, which gives $V_t = \frac{\eta\sigma^2}{4}I$. Substituting $V_t$ into the dynamics equation 12 we have the following effective dynamics

$$dz_t = -\frac{\eta\sigma^2}{4}\nabla_{\mathcal{M}} \text{Tr}(\tilde{H}(z_t))dt. \tag{13}$$

In the case of a flat manifold, this result above corresponds to the dynamics we derived in the introduction. Effectively, the SGD is minimizing $\frac{\eta\sigma^2}{4}\text{Tr}(H(z))$ on the manifold using a gradient flow. (Note that $\text{Tr}(\tilde{H}(z)) = \text{Tr}(H(z))$ for $z \in \mathcal{M}$).

For another case, if we assume the noise in $T_z\mathcal{M}^\perp$ is isotropic with a constant magnitude, we have $\tilde{D}_{22}(z) = \sigma I$ for some $\sigma$. Hence, we have $V_t = \frac{\eta\sigma^2}{4}\tilde{H}^{-1}(z_t)$, and the manifold dynamics becomes

$$dz_t = -\frac{\eta\sigma^2}{4}\nabla_{\mathcal{M}} \text{Tr}(\log \tilde{H}(z_t))dt. \tag{14}$$

Effectively, the SGD is minimizing $\frac{\eta\sigma^2}{4}\text{Tr}(\log \tilde{H}(z))$ on the manifold with a gradient flow.

---

[1]Although 11 is a dynamics for $\mathbf{x} \in \mathbb{R}^k$, it can be understood as a dynamics for $z$ on $T_{z_0}\mathcal{M}$, in which the $y$ component is always 0.

**Remark 1.** *The manifold dynamics we derive is similar to that studied in Li et al. (2021b). Instead of providing a rigorous proof, our main contribution is to give a simple and intuitive quasistatic approach to derive the manifold dynamics. Our methods can be applied to a wide class of noise models, and also can be applied to other optimizers such as SGD with momentum (See Section 2.3) and Adam (See Section 5).*

## 2.3 EXTENDING ANALYSIS TO SGD WITH MOMENTUM

The quasistatic approach we take can be extended to derive the effective manifold dynamics for SGDm. We consider the following SGDm scheme on the same loss function $f$ studied above:

$$\mathbf{m}_{k+1} = \mu\mathbf{m}_k - \eta\nabla f(\mathbf{z}_k), \quad \mathbf{z}_{k+1} = \mathbf{z}_k + \mathbf{m}_{k+1}, \tag{15}$$

where $\eta$ is the learning rate, $\mu \in [0,1)$ is the momentum factor, and $\mathbf{m}$ is the momentum. By the derivation in Li et al. (2017), we consider the following SDE system that approximates 15:

$$d\mathbf{m}_t = -\big(\frac{1-\mu}{\eta}\mathbf{m}_t + \nabla f(\mathbf{z}_t)\big)dt + \sqrt{\eta}D(\mathcal{P}\mathbf{z}_t)dW_t, \quad d\mathbf{z}_t = \frac{1}{\eta}\mathbf{m}_t dt. \tag{16}$$

By the discussion for SGD, the manifold dynamics can be obtained by assuming that $\mathcal{M}$ is flat, and applying a quasistatic analysis on a decomposition of tangent and normal components. We put the details in Appendix A. The resulting manifold dynamics on a general manifold is

$$d\mathbf{m}_t = -\left(\frac{1-\mu}{\eta}\mathbf{m}_t + \sum_{i,j=1}^{n-k}(V_t^{\text{sgdm}})_{ij}\nabla_{\mathcal{M}}(\tilde{H}(\mathbf{z}_t)_{ij})\right)dt + \sqrt{\eta}\tilde{D}_{11}(\mathbf{z}_t)dW_t, \quad d\mathbf{z}_t = \frac{1}{\eta}\mathbf{m}_t dt, \tag{17}$$

where the form and derivation of $V_t^{\text{sgdm}}$ are given in Appendix A (equation 39). Note that when $\eta$ is small, equation 39 is close to $\tilde{H}V + V\tilde{H} = \frac{\eta}{2(1-\mu)}\tilde{D}_{22}\tilde{D}_{22}^T$. Let $\tilde{V}^{\text{sgdm}}$ be the solution of this equation. We have $V^{\text{sgdm}} \approx \tilde{V}^{\text{sgdm}} = \frac{V^{\text{sgd}}}{1-\mu}$, where $V^{\text{sgd}}$ is the matrix $V$ for SGD used in previous sections. This shows that the momentum amplifies the flatness driven force by a factor of $1/(1-\mu)$. Besides this acceleration, the momentum scheme itself also accelerates the speed of manifold dynamics. To see this, when there is no noise along the manifold direction, i.e. $\tilde{D}_{11} = 0$, by Kovachki & Stuart (2021), the ODE

$$\dot{\mathbf{z}}_t = -\frac{1}{1-\mu}\sum_{i,j=1}^{n-k}(V_t^{\text{sgdm}})_{ij}\nabla_{\mathcal{M}}(\tilde{H}(\mathbf{z}_t)_{ij}) \tag{18}$$

is a first-order approximation of the manifold dynamics 17 (This approximation assumes the momentum is always at equilibrium). The term $1/(1-\mu)$ is the acceleration brought by the momentum scheme. In this case, compared with SGD, the approximate manifold dynamics for SGDm is

$$\dot{\mathbf{z}}_t = -\frac{1}{(1-\mu)^2}\sum_{i,j=1}^{n-k}(V_t^{\text{sgd}})_{ij}\nabla_{\mathcal{M}}(\tilde{H}(\mathbf{z}_t)_{ij}),$$

which is $1/(1-\mu)^2$ faster than SGD. The full derivation for SGDm are put in Appendix A.

**Examples:** We still consider the example where $\tilde{D}_{11} = 0$ and $\tilde{D}_{22}(\mathbf{z}) = \alpha\sqrt{\tilde{H}(\mathbf{z})}$. In this case, $V^{\text{sgdm}} \approx \tilde{V}^{\text{sgdm}} = \frac{\eta\alpha^2}{4(1-\mu)}I$. Then, the approximate effective dynamics according to equation 18 is

$$\dot{\mathbf{z}}_t = -\frac{\eta\alpha^2}{4(1-\mu)^2}\nabla_{\mathcal{M}}\text{Tr}(\tilde{H}(\mathbf{z}_t)). \tag{19}$$

As a numerical justification for the manifold dynamics we derived for SGDm, for the same function $f(x,y) = (x^2+1)y^2$ tested in Figure 1, we compare the true x-coordinate dynamics, the SGDm-like discretization for equation 17, and the ODE solution of 19. Shown in the left panel of Figure 2, the three dynamics are close for all $\mu$ tested. The results also show that the manifold dynamics get faster for larger $\mu$, as predicted by its expression.

## 3 APPLICATION ON MATRIX FACTORIZATION PROBLEMS

In this section, we consider an objective function $f(U,V) = \|UV^T - M\|_F^2$ with $U \in \mathbb{R}^{m\times p}, V \in \mathbb{R}^{n\times p}$ and an SGD with Hessian noise, i.e. the noise covariance is proportional with the Hessian of $f$.

Let $H(U,V)$ be the Hessian matrix of $f$. Theoretically, it is easy to verify that $\mathrm{Tr}(H(U,V))$ is proportional to $\|U\|_F^2 + \|V\|_F^2$ wherever $UV^T = M$. Therefore, the manifold dynamics of SGD minimizes $\|U\|_F^2 + \|V\|_F^2$ and drives the iterator to the most balanced

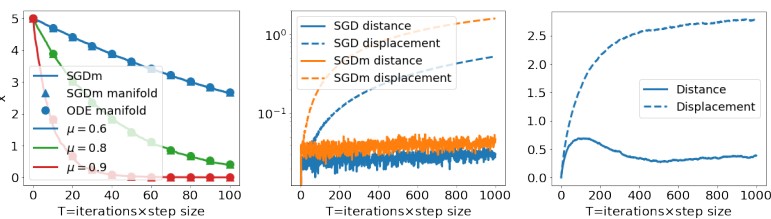

Figure 2: **(left)** SGDm and its manifold dynamics for different $\mu$. **(middle)** The distance between the real optimization dynamics and the manifold dynamics for SGD and SGDm with Hessian noise, compared with the displacement of the optimization dynamics' iterators. **(right)** The distance and displacement curves for average trajectories of SGD when there is noise long the minima manifold.

global minimum, which is also the flattest. Numerically, we take $m = n = p = 5$ and compare the true dynamics with the manifold dynamics. The experiments are initialized from $U_0 = M, V_0 = I$. The middle panel of Figure 2 shows the distance between the true dynamics and the manifold dynamics, as well as the distance traveled by the true dynamics, for SGD and SGDm. On the right panel, we inject an isotropic noise on the tangent space for SGD and do the same comparison. The results show good approximation of the manifold dynamics to the real dynamics. Under the Hessian noise, then, the SGD indeed moves towards minima with smaller Frobenius norm, same as the manifold dynamics.

## 4 LEARNING RATE, BATCH SIZE, AND MOMENTUM

With the manifold dynamics, we can study the impact of hyperparameters on the behavior of optimizers around manifold of minima. Here, we focus on the learning rate $\eta$, batch size $B$, and momentum $\mu$ of the SGDm algorithm. Written in the SDE form, the batch size changes the covariance of the noise by a factor $1/B$. Hence, using equation 16, the SDE with batch size $B$ is

$$d\mathbf{m}_t = -\big(\frac{1-\mu}{\eta}\mathbf{m}_t + \nabla f(\boldsymbol{z}_t)\big)dt + \frac{\sqrt{\eta}}{\sqrt{B}}D(\mathcal{P}\boldsymbol{z}_t)dW_t, \quad d\boldsymbol{z}_t = \frac{1}{\eta}\mathbf{m}_t dt. \tag{20}$$

To focus on the drift dynamics on the manifold and avoid the influence of the noise, we consider Hessian noise which only exists in the normal space of the manifold, i.e. we assume $\tilde{D}_{11} = 0$, $\tilde{D}_{22}(\boldsymbol{z}) = \sigma\sqrt{\tilde{H}(\boldsymbol{z})}$. Then, by equation 18, the first-order ODE on the minima manifold representing the manifold dynamics is

$$\dot{\boldsymbol{z}}_t = -\frac{\eta\sigma^2}{4B(1-\mu)^2}\nabla_{\mathcal{M}}\mathrm{Tr}(\tilde{H}(\boldsymbol{z}_t)). \tag{21}$$

By equation 21, the manifold dynamics takes the same trajectory with different speed for different hyperparameters. Let $\tilde{\boldsymbol{z}}_t$ be the trajectory of $\dot{\boldsymbol{z}}_t = -\frac{\sigma^2}{4}\nabla_{\mathcal{M}}\mathrm{Tr}(\tilde{H}(\boldsymbol{z}_t))$. Considering the discretization, an SGDm with learning rate $\eta$, batch size $B$, and momentum $\mu$ takes $\frac{TB(1-\mu)^2}{\eta^2}$ iterations to solve for $\tilde{\boldsymbol{z}}_t$ until $t = T$. Hence, decreasing the batch size, or increasing the learning rate or momentum factor, can accelerate the (discrete) manifold dynamics. We let $s(\eta, \mu, B) := \frac{\eta^2}{B(1-\mu)^2}$ be the speed factor for the dynamics. The experiment results in the left panel of Figure 3 justify that the speed factor indeed controls the dynamics' speed.

**Implications in practical cases:** For the training process of over-parameterized neural networks, the exploration around the minima manifold is an important source of implicit regularization. The driven force of the movement along the manifold is still the change of flatness. However, in this more complicated case, the discussion above may face two problems: (1) The curvature in the directions perpendicular with the manifold may not be quadratic. Which leads to different manifold trajectory if the range of oscillation is different. (i.e. if the iterator oscillates in a larger range, the flatness driven force may change its direction.) (2) The manifold dynamics is just a first-order approximation of the true dynamics, which may not be accurate for a long time period. The second problem is intrinsic to all studies that use continuous dynamics to approximate discrete dynamics. It will not impose

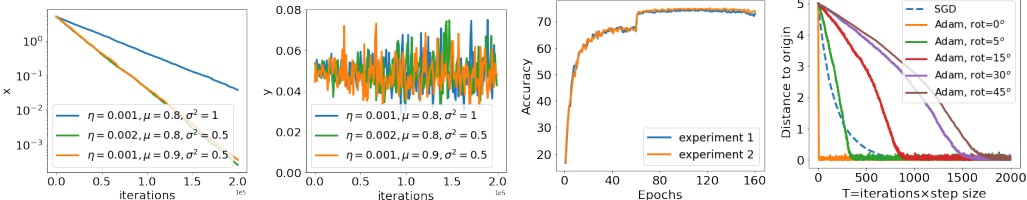

Figure 3: **(Left)** The x-coordinate dynamics of three experiments for SGD with different hyperparameters, on function $f(x, y) = (1 + x^2)y^2$ with Hessian noise. Two experiments have the same $s(\eta, \mu, B)$, while the other one has a smaller $s(\eta, \mu, B)$. Here the role of $B$ is played by $1/\sigma^2$. **(Middle left)** The moving average of y-magnitude for the three experiments shown in the left panel. The three experiments have the same $r(\eta, \mu, B)$, hence the y-magnitudes are on the same order. **(Middle right)** The test accuracy curves of two neural network runs with different hyperparameters but the same $s$ and $r$. Experiments are conducted on ResNet18 and CIFAR100 dataset. **(Right)** The distance between the manifold projection and the origin of Adam iterators for loss function equation 23 with different $\theta$, compared with that for SGD.

a serious problem as long as the curvature on and around the minima manifold does not change drastically. The first problem motivates us to find ways to accelerate the manifold dynamics without changing the range of oscillation. By the discussion in Section 2.3, the range of the oscillation is given by $V^{\text{sgdm}}$, which is $\frac{\eta\alpha^2}{4B(1-\mu)}I$. Let $r(\eta, \mu, B) := \frac{\eta}{B(1-\mu)}$ be the "range factor". Then, Combining the discussions above, we want to increase the speed factor $s(\eta, \mu, B)$ while keeping the range factor $r(\eta, \mu, B)$ fixed. Since the ratio $\eta/(1-\mu)$ appears in both factors, to increase the dynamics speed without changing the oscillation range we must change $B$. Concretely, if we pick $B' = cB$ and $\eta', \mu'$ such that $\frac{\eta'}{1-\mu'} = c\frac{\eta}{1-\mu}$, then

$$\frac{\eta'}{B'(1-\mu')} = \frac{\eta}{B(1-\mu)}, \qquad \frac{\eta'^2}{B'(1-\mu')^2} = c\frac{\eta^2}{B(1-\mu)^2}, \tag{22}$$

i.e. the range factor is not changed while the speed factor is multiplied by $c$. In the second panel of Figure 3, we show for synthetic problem that $r(\eta, \mu, B)$ is indeed proportional to the range of oscillation. In the third panel, we show for neural networks that two experiments with the same speed and range factors but different hyperparameters indeed follow the same test loss curve with respect to epochs. On the other hand, if the oscillation range is not kept, the training trajectories may go to different solutions (with different training and testing error).

**Remark 2.** *By equation 22, increasing the batch size while changing other hyperparameters accordingly can accelerate the training speed without changing the training trajectory. However, this acceleration happens on the level of number of iterations. Since the batch size changes accordingly, the number of samples used during the training period does not change. This means we are training for less iterations, but the same number of epochs. This is shown in the third panel of Figure 3. Nevertheless, this usually saves time because training one big batch is faster than training several small batches with the same number of total samples. Therefore, our results reveals a theoretical mechanism underpinning the empirical benefit of large batch training You et al. (2019; 2017); Hoffer et al. (2017); Geiping et al. (2021).*

## 5 ADAPTIVE GRADIENT METHODS AND ROTATIONAL INVARIANCE

We can also study the manifold dynamics of adaptive gradient methods. We start with experiments which show that the manifold dynamics of Adam changes according to the direction of the manifold. This is due to the fact that adaptive gradient methods are not rotational invariant. We consider the loss function,

$$f(x, y) = (x\sin\theta + y\cos\theta)^2((x\cos\theta - y\sin\theta)^2 + 1), \tag{23}$$

which is the counterclockwise rotation of $(x^2 + 1)y^2$ by $\theta^2$. The minima manifold of $f$ is the line $x\sin\theta + y\cos\theta = 0$. We run Adam Kingma & Ba (2014) on $f$ with different $\theta$ with Hessian noise. The right panel of Figure 3 compares the dynamics projected onto the minima manifold. The results

---

[2]Our analysis works for any loss function with the form $f(x, y) = (x\sin\theta + y\cos\theta)^2 h(x\cos\theta - y\sin\theta)$.

show that Adam moves very fast along the manifold when the manifold aligns well with an axis. When the manifold does not align with an axis, Adam moves much slower, sometimes even slower than a plain SGD. This is because in Adam the adaptive learning rate is computed for each coordinate, and hence when the manifold direction is close to an axis, the learning rate along the manifold can be drastically increased due to the small gradient on this corresponding axis direction. Otherwise, all axis directions have big gradients due to the oscillation and the learning rate along the manifold is not increased in a desirable way.

Using the SDE approximation for Adam recently derived in Malladi et al. (2022), we can derive and compare the manifold dynamics for Adam on the loss equation 23 for different $\theta$. Consider an Adam algorithm with hyperparameters $(\beta_1, \beta_2, \eta, \epsilon)$, where $\beta_1$ and $\beta_2$ are momentum coefficients for the first and second order moments, respectively, $\eta$ is the learning rate, and $\epsilon$ is the small number that prevents division by zero Kingma & Ba (2014). By Malladi et al. (2022), let $\Sigma$ be the gradient noise covariance matrix depending on the parameters, and $\sigma$ be a additional noise strength (i.e. the real noise covariance is $\sigma\Sigma$), define $\sigma_0 = \sigma\eta$, $\epsilon_0 = \epsilon\eta$, $c_1 = (1 - \beta_1)/\eta^2$, $c_2 = (1 - \beta_2)/\eta^2$, $\gamma_1(t) = 1 - e^{-c_1 t}$, and $\gamma_2(t) = 1 - e^{-c_2 t}$, then the Adam trajectory is approximated by the SDE:

$$d\mathbf{x}_t = -\frac{\sqrt{\gamma_2(t)}}{\gamma_1(t)} P_t^{-1} \mathbf{m}_t dt, \qquad d\mathbf{m}_t = c_1(\nabla f(\mathbf{x}_t) - \mathbf{m}_t)dt + \sigma_0 c_1 \Sigma^{1/2}(\mathbf{x}_t)dW_t, \qquad (24)$$

$$d\boldsymbol{u}_t = c_2(\text{diag}(\Sigma(\mathbf{x}_t)) - \boldsymbol{u}_t)dt, \qquad P_t = \sigma_0 \text{diag}(\boldsymbol{u}_t)^{1/2} + \epsilon_0 \sqrt{\gamma_2(t)}I.$$

Here, $f$ is the loss function, $\mathbf{x}$ is the parameter, and $W_t$ is a Brownian motion. The time scale of the SDE above is $t = k\eta^2$, which is different from the usual time scale $t = k\eta$ studies for other optimization algorithms.

Using the quasistatic approach, in Appendix B we derive the approximate manifold dynamics for two cases: (1) $\theta = 0$, in which the minima manifold aligns with one axis, and (2) $\theta = \pi/4$, in which the angles between the minima manifold and coordinate axes are maximized. After some approximations which are detailed in Appendix B (such as $\gamma_1(t) = \gamma_2(t) = 1$ which happens when $t$ is large), we have the following effective dynamics on the minima manifold:

$$\theta = 0 : \quad dm_{x,t} = c_1 \left( \frac{\sigma_0 h'(x_t)}{4\sqrt{h(x_t)}} - m_{x,t} \right) dt, \quad dx_t = -\frac{m_{x,t}}{\epsilon_0} dt. \qquad (25)$$

$$\theta = \frac{\pi}{4} : \quad dm_{x,t} = c_1 \left( \frac{\sigma_0 h'(x_t)}{2\sqrt{2h(x_t)}} - m_{x,t} \right) dt, \quad dx_t = -\frac{\sqrt{2}m_{x,t}}{\sigma_0 \sqrt{h(x_t)}} dt. \qquad (26)$$

Here, $x$ is the coordinate along the manifold direction, and $m$ is a corresponding momentum. The SDEs equation 25 and equation 26 show the difference of the manifold dynamics for different $\theta$. When $\theta = 0$, the $x$ dynamics is very fast, because of the $\epsilon_0$ on the denominator. When $\theta = \pi/4$, instead, the $x$ dynamics is slower. If we further make a first-order approximation of the dynamics by assuming the momentum is always at equilibrium, like we did for SGDm in equation 18, we have the following manifold dynamics:

$$\theta = 0 : \quad \dot{x} = -\frac{\sigma_0 h'(x)}{4\epsilon_0 \sqrt{h(x)}}, \qquad \theta = \frac{\pi}{4} : \quad \dot{x} = -\frac{h'(x)}{2h(x)}. \qquad (27)$$

Here we see that when $\theta = 0$ we get a gradient flow minimizing $\sqrt{h(x)}$ on the minima manifold, while when $\theta = \pi/4$ we get a gradient flow minimizing $\ln h(x)$. The former dynamics is much faster due to the $\epsilon_0$ on the denominator. For the detail of the analysis please see Appendix B. When the dimension of the parameter space is high, it is hard for the minima manifold to align well with coordinate directions. Hence, the exploration of Adam (as well as other adaptive gradient methods) on the minima manifold is slower than SGD and SGDm. This may be one reason that Adam does not generalize as good as SGD in many cases Keskar & Socher (2017); Wilson et al. (2017).

**Remark 3.** *Unlike the effective dynamics for SGD and SGDm, the dynamics in equation 27 do not depend on the learning rate $\eta$. This is because in the SDE approximation the time scale is $t = k\eta^2$. Therefore, a $\eta$ factor will appear if we transform the time scale to $t = k\eta$.*

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

## A    SGD WITH MOMENTUM

Consider the following SDE approximation for SGDm:

$$d\mathbf{m}_t = -\big(\frac{1-\mu}{\eta}\mathbf{m}_t + \nabla f(\mathbf{z}_t)\big)dt + \sqrt{\eta}D(\mathcal{P}\mathbf{z}_t)dW_t, \quad d\mathbf{z}_t = \frac{1}{\eta}\mathbf{m}_t dt. \tag{28}$$

By the discussion for SGD, the manifold dynamics can be obtained by assuming that $\mathcal{M}$ is flat. The result for non-flat manifold is different only by a gradient taken on the manifold. In the flat case, still

letting $\boldsymbol{z} = [\mathbf{x}^T, \boldsymbol{y}^T]^T$ with $\mathbf{x}$ in the tangent space and $\boldsymbol{y}$ in the normal space, the dynamics 28 can be written as an SDE system for $\mathbf{x}$ and $\boldsymbol{y}$.

$$dm_{x,t} = -\left(\frac{1-\mu}{\eta}m_{x,t} + \boldsymbol{v}_t^T \partial_{\boldsymbol{u}} \tilde{H}(\mathbf{x}_t)\boldsymbol{y}_t\right) dt + \sqrt{\eta}\tilde{D}_{11}(\mathbf{x}_t)dW_t^{(1)}, \quad d\mathbf{x}_t = \frac{1}{\eta}m_{x,t}dt,$$

$$dm_{y,t} = -\left(\frac{1-\mu}{\eta}m_{y,t} + 2\tilde{H}(\mathbf{x}_t)\boldsymbol{y}_t\right) dt + \sqrt{\eta}\tilde{D}_{22}(\mathbf{x}_t)dW_t^{(2)}, \qquad d\boldsymbol{y}_t = \frac{1}{\eta}m_{y,t}dt. \quad (29)$$

Here, $\mathbf{m}_{x,t}$ and $\mathbf{m}_{y,t}$ denote the momentum for $\mathbf{x}$ and $\boldsymbol{y}$ components, respectively. Again, since the $\boldsymbol{y}$ dynamics is faster than the $\mathbf{x}$ dynamics, we take a quasistatic approach by assuming $\boldsymbol{y}_t$ is always at the equilibrium given $\mathbf{x}_t$, and taking expectation on $\boldsymbol{y}_t$ in the $\mathbf{x}$ dynamics. Note that the two equations for $\mathbf{m}_{y,t}$ and $\boldsymbol{y}_t$ form a linear system of SDEs, we can still compute the first and second moments of $\boldsymbol{y}_t$ at equilibrium and substitute the results into the equations for $\mathbf{m}_{\mathbf{x},t}$ and $\mathbf{x}$, which gives the following effective dynamics for $\mathbf{x}$

$$d\mathbf{m}_{\mathbf{x},t} = -\left(\frac{1-\mu}{\eta}\mathbf{m}_{\mathbf{x},t} + \sum_{i,j=1}^{n-k}(V_t^{\text{sgdm}})_{ij}\nabla_{\mathbf{x}}(\tilde{H}(\mathbf{x}_t)_{ij})\right) dt + \sqrt{\eta}\tilde{D}_{11}(\mathbf{x}_t)dW_t, \ \ d\mathbf{x}_t = \frac{1}{\eta}\mathbf{m}_{\mathbf{x},t}dt,$$

where the form and derivation of $V_t^{\text{sgdm}}$ are given in the next subsection. Finally, replacing $\mathbf{x}$ by $\boldsymbol{z}$ on $\mathcal{M}$ and consider gradients on $\mathcal{M}$, we have the following effective dynamics on general minima manifold:

$$d\mathbf{m}_t = -\left(\frac{1-\mu}{\eta}\mathbf{m}_t + \sum_{i,j=1}^{n-k}(V_t^{\text{sgdm}})_{ij}\nabla_{\mathcal{M}}(\tilde{H}(\boldsymbol{z}_t)_{ij})\right) dt + \sqrt{\eta}\tilde{D}_{11}(\boldsymbol{z}_t)dW_t, \ \ d\boldsymbol{z}_t = \frac{1}{\eta}\mathbf{m}_tdt,$$

$$(30)$$

## A.1 THE DERIVATION FOR $V^{\text{sgdm}}$

In this section, we derive $V_{\text{sgdm}}$ from the SDE for SGDm equation 16. Assume $\mathbf{x}_t$ and $\mathbf{m}_{\mathbf{x},t}$ is fixed, we search for the equilibrium of the following system of $\boldsymbol{y}$ and $\mathbf{m}_{\boldsymbol{y}}$:

$$d\mathbf{m}_{\boldsymbol{y},\tau} = -\left(\frac{1-\mu}{\eta}\mathbf{m}_{\boldsymbol{y},\tau} + 2\tilde{H}(\mathbf{x})\boldsymbol{y}_\tau\right) d\tau + \sqrt{\eta}\tilde{D}_{22}(\mathbf{x}_t)dW_\tau^{(2)},$$

$$d\boldsymbol{y}_t = \tau = \frac{1}{\eta}\mathbf{m}_{\boldsymbol{y},\tau}d\tau.$$

The SDE system above is linear. Let $\boldsymbol{u}_\tau = \begin{bmatrix} \mathbf{m}_{\boldsymbol{y},\tau} \\ \boldsymbol{y}_\tau \end{bmatrix}$, the SDE can be written as

$$d\boldsymbol{u}_\tau = A\boldsymbol{u}_\tau d\tau + \sqrt{\eta}DdB_\tau, \qquad (31)$$

where we have

$$A = \begin{bmatrix} -\frac{1-\mu}{\eta}I & -2\tilde{H}(\mathbf{x}_t) \\ \frac{1}{\eta}I & 0 \end{bmatrix} \in \mathbb{R}^{2(d-k)\times 2(d0k)}, \qquad D = \begin{bmatrix} 0 & \tilde{D}_{22} \\ 0 & 0 \end{bmatrix} \in \mathbb{R}^{2(d-k)\times d},$$

and $B_\tau$ is a Brownian motion.

Let $C_\tau$ be the second moment matrix $\mathbb{E}\boldsymbol{u}_\tau\boldsymbol{u}_\tau^T$. By Herzog (2013), $C_\tau$ satisfies the ODE

$$\frac{d}{d\tau}C_\tau = AC_\tau + C_\tau A^T + \eta DD^T.$$

Therefore, taking $\tau \to \infty$, let $C_\infty = \lim_{\tau\to\infty} C_\tau$ be the moment matrix at equilibrium, we have

$$AC_\infty + C_\infty A^T = -\eta DD^T. \qquad (32)$$

By the definition of $\boldsymbol{u}$, we have

$$C_\infty = \begin{bmatrix} \mathbb{E}\mathbf{m}_{\boldsymbol{y},\infty}\mathbf{m}_{\boldsymbol{y},\infty}^T & \mathbb{E}\mathbf{m}_{\boldsymbol{y},\infty}\boldsymbol{y}_\infty^T \\ \mathbb{E}\boldsymbol{y}_\infty\mathbf{m}_{\boldsymbol{y},\infty}^T & \mathbb{E}\boldsymbol{y}_\infty\boldsymbol{y}_\infty^T \end{bmatrix}.$$

We are interested in the $\mathbb{E}\boldsymbol{y}_\infty \boldsymbol{y}_\infty^T$ in the above matrix. By the symmetry of $C_\infty$, let

$$C_\infty = \left[\begin{array}{cc} C_1 & C_2 \\ C_2^T & C_3 \end{array}\right].$$

Then, we want to derive $C_3$. Substituting the blockwise $C_\infty$ into equation 32, we have

$$\left[\begin{array}{cc} -\frac{1-\mu}{\eta}I & -2\tilde{H}(\mathbf{x}) \\ \frac{1}{\eta}I & 0 \end{array}\right]\left[\begin{array}{cc} C_1 & C_2 \\ C_2^T & C_3 \end{array}\right] + \left[\begin{array}{cc} C_1 & C_2 \\ C_2^T & C_3 \end{array}\right]\left[\begin{array}{cc} -\frac{1-\mu}{\eta}I & \frac{1}{\eta}I \\ -2\tilde{H}(\mathbf{x}) & 0 \end{array}\right] = -\eta\left[\begin{array}{cc} \tilde{D}_{22}\tilde{D}_{22}^T & 0 \\ 0 & 0 \end{array}\right],$$

which gives

$$-\frac{2(1-\mu)}{\eta}C_1 - 2\big(\tilde{H}(\mathbf{x})C_2^T + C_2\tilde{H}(\mathbf{x})\big) = -\eta\tilde{D}_{22}\tilde{D}_{22}^T, \tag{33}$$

$$\frac{1}{\eta}C_1 - \frac{1-\mu}{\eta}C_2 - 2\tilde{H}(\mathbf{x})C_3 = 0 \tag{34}$$

$$\frac{1}{\eta}C_2 + \frac{1}{\eta}C_2^T = 0. \tag{35}$$

By equation 35, $C_2$ is skew symmetric. By definition, $C_1$ and $C_3$ are symmetric. Hence, adding equation 34 with its transpose, we obtain

$$\frac{1}{\eta}C_1 - \big(\tilde{H}(\mathbf{x})C_3 + C_3\tilde{H}(\mathbf{x})\big) = 0,$$

which gives

$$C_1 = \eta\big(\tilde{H}(\mathbf{x})C_3 + C_3\tilde{H}(\mathbf{x})\big). \tag{36}$$

Substituting into equation 35, we have

$$C_2 = \frac{\eta}{1-\mu}\big(C_3\tilde{H}(\mathbf{x}) - \tilde{H}(\mathbf{x})C_3\big). \tag{37}$$

Plugging equation 36 and equation 37 into equation 33, we have

$$2(1-\mu)\big(\tilde{H}(\mathbf{x})C_3 + C_3\tilde{H}(\mathbf{x})\big) + \frac{2\eta}{1-\mu}\big(C_3\tilde{H}(\mathbf{x})^2 + \tilde{H}(\mathbf{x})^2 C_3 - 2\tilde{H}(\mathbf{x})C_3\tilde{H}(\mathbf{x})\big)$$

$$= \eta\tilde{D}_{22}\tilde{D}_{22}^T. \tag{38}$$

There, denote $\tilde{H} = \tilde{H}(\mathbf{x})$, $V_{\text{sgdm}}$ is the solution of

$$\tilde{H}V + V\tilde{H} + \frac{\eta}{(1-\mu)^2}\left(V\tilde{H}^2 + \tilde{H}^2 V - 2\tilde{H}V\tilde{H}\right) = \frac{\eta}{2(1-\mu)}\tilde{D}_{22}\tilde{D}_{22}^T. \tag{39}$$

## B  ADAM AND RMSPROP

In this section, we derive the effective dynamics equation 25 and equation 26 for Adam, on the 2-D problem equation 23:

$$f(x, y) = (x\sin\theta + y\cos\theta)((x\cos\theta - y\sin\theta)^2 + 1).$$

We first describe the relation between two coordinate systems. Let $xOy$ be the coordinate system on which the loss function equation 23 is defined, and Adam is conducted. Let $x'Oy'$ be the coordinate system obtained by rotating $xOy$ counterclockwise by $\theta$. Then, the $x'$ axis aligns with the direction of the minima manifold of $f$. In this coordinate system, $f$ has the form:

$$f(x, y) = ((x')^2 + 1)(y')^2.$$

Let $R_\theta$ be the rotation matrix counterclockwise by $\theta$, i.e. $R_\theta = \left[\begin{array}{cc} \cos\theta & -\sin\theta \\ \sin\theta & \cos\theta \end{array}\right]$. Then, for any vector $\mathbf{x}$ in $xOy$, let $\mathbf{x}'$ be its coordinate in $x'Oy'$, we have $\mathbf{x}' = R_{-\theta}\mathbf{x}$, and $\mathbf{x} = R_\theta\mathbf{x}'$.

Recall the SDE for Adam derived in Malladi et al. (2022),

$$d\mathbf{x}_t = -\frac{\sqrt{\gamma_2(t)}}{\gamma_1(t)} P_t^{-1} \mathbf{m}_t dt, \qquad d\mathbf{m}_t = c_1(\nabla f(\mathbf{x}_t) - \mathbf{m}_t)dt + \sigma_0 c_1 \Sigma^{1/2}(\mathbf{x}_t)dW_t, \qquad (40)$$

$$d\boldsymbol{u}_t = c_2(\text{diag}(\Sigma(\mathbf{x}_t)) - \boldsymbol{u}_t)dt, \qquad P_t = \sigma_0 \text{diag}(\boldsymbol{u}_t)^{1/2} + \epsilon_0 \sqrt{\gamma_2(t)}I,$$

where $\mathbf{x}$ is the parameter vector, $\mathbf{m}$ is the first momentum vector, $\boldsymbol{u}$ is the second momentum vector, and $\Sigma$ is the noise covariance. All these quantities are defined in the coordinate system $xOy$. Let $\mathbf{x}'$, $\mathbf{m}'$, $\Sigma'$ be the counterparts of $\mathbf{x}$, $\mathbf{m}$, $\Sigma$ in $x'Oy'$, then

$$\mathbf{x}' = R_{-\theta}\mathbf{x}, \quad \mathbf{m}' = R_{-\theta}\mathbf{m}, \quad \Sigma' = R_{-\theta}\Sigma R_\theta. \qquad (41)$$

We do not consider $\boldsymbol{u}'$ as $\boldsymbol{u}$ in $x'Oy'$, because $\boldsymbol{u}$ is not rotationally invariant. This is the reason that Adam has different effective dynamics on the minima manifold for different $\theta$.

By the relations equation 41, the SDE equation 40 can be written as

$$d\mathbf{x}'_t = -\frac{\sqrt{\gamma_2(t)}}{\gamma_1(t)} R_{-\theta} P_t^{-1} R_\theta \mathbf{m}'_t dt, \qquad d\mathbf{m}'_t = c_1(R_{-\theta}\nabla f(\mathbf{x}_t) - \mathbf{m}'_t)dt + \sigma_0 c_1 \Sigma'^{1/2}(\mathbf{x}_t)dB_t,$$

$$(42)$$

$$d\boldsymbol{u}_t = c_2(\text{diag}(\Sigma(\mathbf{x}_t)) - \boldsymbol{u}_t)dt, \qquad P_t = \sigma_0 \text{diag}(\boldsymbol{u}_t)^{1/2} + \epsilon_0 \sqrt{\gamma_2(t)}I,$$

where $B_t = R_{-\theta}W_t$ is a Brownian motion in the $x'Oy'$ system. Note that $R_{-\theta}\nabla f(\mathbf{x})$ is the gradient of $f$ in the $x'Oy'$ system, letting $\mathbf{x} = [x, y]^T$ and $\mathbf{x}' = [x', y']^T$, we have

$$R_{-\theta}\nabla f(\mathbf{x}) = \begin{bmatrix} 2(x')y'^2 \\ 2(x'^2 + 1)y' \end{bmatrix}.$$

From now on, we denote $h(x) = x^2 + 1$. Then

$$R_{-\theta}\nabla f(\mathbf{x}) = \begin{bmatrix} h'(x')y'^2 \\ 2h(x')y' \end{bmatrix}.$$

Our analysis actually works for any positive and differentiable function $h$. Under the Hessian noise assumption, we take

$$\Sigma'(\mathbf{x}) = \begin{bmatrix} 0 & 0 \\ 0 & h(x') \end{bmatrix},$$

then for $\Sigma$ we have

$$\Sigma(\mathbf{x}) = R_\theta \Sigma'(\mathbf{x})R_{-\theta} = h(x') \begin{bmatrix} \sin^2\theta & -\sin\theta\cos\theta \\ -\sin\theta\cos\theta & \cos^2\theta \end{bmatrix}.$$

We do not add a $\sigma$ before $\sqrt{h(x')}$ because in the derivation of the SDE equation 40 a strength factor $\sigma$ is included into $\sigma_0$.

By the discussion on $R_{-\theta}\nabla f(\mathbf{x})$ and $\Sigma'(\mathbf{x})$, letting $\mathbf{m}' = [m_{x'}, m_{y'}]^T$, $\boldsymbol{u} = [u, v]^T$, we can write equation 42 into the following system of $x', y', m_{x'}, m_{y'}, u, v$:

$$\begin{bmatrix} dx'_t \\ dy'_t \end{bmatrix} = -\frac{\sqrt{\gamma_2(t)}}{\gamma_1(t)} R_{-\theta} P_t^{-1} R_\theta \begin{bmatrix} m_{x',t}dt \\ m_{y',t}dt \end{bmatrix}$$

$$dm_{x',t} = c_1(h'(x'_t)y_t'^2 - m_{x',t})dt$$

$$dm_{y',t} = c_1(2h(x'_t)y'_t - m_{y',t})dt + \sigma_0 c_1 \sqrt{h(x'_t)}dB_t$$

$$du_t = c_2(h(x'_t)\sin^2\theta - u_t)dt$$

$$dv_t = c_2(h(x'_t)\cos^2\theta - v_t)dt$$

$$P_t = \sigma_0 \begin{bmatrix} u_t & 0 \\ 0 & v_t \end{bmatrix}^{1/2} + \epsilon_0 \sqrt{\gamma_2(t)}I. \qquad (43)$$

In equation 43, $B_t$ is a 1-D Brownian motion.

Next, we consider two cases: $\theta = 0$ and $\theta = \frac{\pi}{4}$.

**Case $\theta = 0$.** When $\theta = 0$, we have $R_\theta = R_{-\theta} = I$. Also, since

$$P_t = \sigma_0 \begin{bmatrix} u_t & 0 \\ 0 & v_t \end{bmatrix}^{1/2} + \epsilon_0 \sqrt{\gamma_2(t)} = \begin{bmatrix} \sigma_0\sqrt{u_t} + \epsilon_0\sqrt{\gamma_2(t)} & 0 \\ 0 & \sigma_0\sqrt{v_t} + \epsilon_0\sqrt{\gamma_2(t)} \end{bmatrix},$$

we have

$$P_t^{-1} = \begin{bmatrix} \frac{1}{\sigma_0\sqrt{u_t} + \epsilon_0\sqrt{\gamma_2(t)}} & 0 \\ 0 & \frac{1}{\sigma_0\sqrt{v_t} + \epsilon_0\sqrt{\gamma_2(t)}} \end{bmatrix}.$$

Therefore, equation 43 can be written as

$$\begin{aligned}
dx_t' &= -\frac{\sqrt{\gamma_2(t)}}{\gamma_1(t)} \frac{m_{x',t}}{\sigma_0\sqrt{u_t} + \epsilon_0\sqrt{\gamma_2(t)}} dt \\
dy_t' &= -\frac{\sqrt{\gamma_2(t)}}{\gamma_1(t)} \frac{m_{y',t}}{\sigma_0\sqrt{v_t} + \epsilon_0\sqrt{\gamma_2(t)}} dt \\
dm_{x',t} &= c_1(h'(x_t')y_t'^2 - m_{x',t})dt \\
dm_{y',t} &= c_1(2h(x_t')y_t' - m_{y',t})dt + \sigma_0 c_1\sqrt{h(x_t')}dB_t \\
du_t &= -c_2 u_t dt \\
dv_t &= c_2(h(x_t') - v_t)dt.
\end{aligned} \tag{44}$$

Using the quasistatic approach, we assume the dynamics of $y'$ and $m_{y'}$ is at equilibrium at any fixed $t$. Hence, fixing $x_t'$, we consider the system

$$\begin{aligned}
dy_\tau' &= -\frac{\sqrt{\gamma_2(t)}}{\gamma_1(t)} \frac{m_{y',\tau}}{\sigma_0\sqrt{v_t} + \epsilon_0\sqrt{\gamma_2(t)}} d\tau \\
dm_{y',\tau} &= c_1(2h(x_t')y_\tau' - m_{y',\tau})d\tau + \sigma_0 c_1\sqrt{h(x_t')}dB_\tau
\end{aligned}$$

and compute $\lim_{\tau\to\infty} \mathbb{E}y_\tau'^2$. This is a linear SDE system of $y'$ and $m_{y'}$, by techniques in Herzog (2013) we have

$$\lim_{\tau\to\infty} \mathbb{E}y_\tau'^2 = \frac{\sigma_0^2\sqrt{\gamma_2(t)}}{4\gamma_1(t)(\sigma_0\sqrt{v_t} + \epsilon_0\sqrt{\gamma_2(t)})}. \tag{45}$$

Substituting equation 45 into equation 44, we have the following effective dynamics only on the minima manifold:

$$\begin{aligned}
dx_t' &= -\frac{\sqrt{\gamma_2(t)}}{\gamma_1(t)} \frac{m_{x',t}}{\sigma_0\sqrt{u_t} + \epsilon_0\sqrt{\gamma_2(t)}} dt \\
dm_{x',t} &= c_1\left(\frac{\sigma_0^2\sqrt{\gamma_2(t)}h'(x_t')}{4\gamma_1(t)(\sigma_0\sqrt{v_t} + \epsilon_0\sqrt{\gamma_2(t)})} - m_{x',t}\right)dt \\
du_t &= -c_2 u_t dt \\
dv_t &= c_2(h(x_t') - v_t)dt.
\end{aligned} \tag{46}$$

Next, we try to make the manifold dynamics equation 46 simpler by doing some approximations. First, solving the ODEs for $u_t$ and $v_t$, we get

$$u_t = u_0 e^{-c_2 t}, \qquad v_t = v_0 e^{-c_2 t} + c_2 \int_0^t e^{c_2(s-t)}h(x_s')ds.$$

We first assume $t$ is big enough, such that $e^{-t}$ is close to $0$. Then, we can take

$$u_t = 0, \qquad v_t = c_2 \int_0^t e^{c_2(s-t)}h(x_s')ds,$$

and also $\gamma_1(t) = \gamma_2(t) = 1$. Moreover, since the dynamics of $x'$ is slow, we can assume the change of $h(x_s)$ is slow compared with $e^{c_2 s}$. In this case, we have

$$c_2 \int_0^t e^{c_2(s-t)}h(x_s')ds \approx h(x_t').$$

Hence, we can approximate $u_t$ and $v_t$ by

$$u_t = 0, \qquad v_t = h(x'_t). \tag{47}$$

Substituting equation 47 into equation 46, and taking $\gamma_1(t) = \gamma_2(t) = 1$, we get the following approximate manifold dynamics:

$$dx'_t = -\frac{m_{x',t}}{\epsilon_0} dt$$

$$dm_{x',t} = c_1 \left( \frac{\sigma_0^2 h'(x'_t)}{4\sigma_0 \sqrt{h(x'_t)} + 4\epsilon_0} - m_{x',t} \right) dt. \tag{48}$$

Finally, in the denominator of the dynamics of $m_{x'}$, the $\epsilon_0$ term is usually small compared with the $\sigma_0 \sqrt{h(x'_t)}$ term. Hence, we can take $4\sigma_0 \sqrt{h(x'_t)} + 4\epsilon_0 \approx 4\sigma_0 \sqrt{h(x'_t)}$ and write the following approximate manifold dynamics:

$$dx'_t = -\frac{m_{x',t}}{\epsilon_0} dt$$

$$dm_{x',t} = c_1 \left( \frac{\sigma_0 h'(x'_t)}{4\sqrt{h(x'_t)}} - m_{x',t} \right) dt. \tag{49}$$

Note that $\frac{h'(x)}{\sqrt{h(x)}} = (2\sqrt{h(x)})'$, the dynamics equation 49 can be understood as a gradient flow with momentum that minimizes $\sqrt{h}$ on the minima manifold. In the region that $h$ is close to linear, we can suppose

$$m_{x',t} = \frac{\sigma_0 h'(x'_t)}{4\sqrt{h(x'_t)}},$$

and write down the following first-order approximation of the dynamics:

$$\frac{dx'_t}{dt} = \frac{\sigma_0 h'(x'_t)}{4\epsilon_0 \sqrt{h(x'_t)}}. \tag{50}$$

Note that due to the time scale chosen in the SDE equation 40, one time unit of the manifold dynamics that we derive corresponds to $1/\eta^2$ steps in the SGD trajectory, i.e. each SGD step corresponds to a time period of $\eta^2$. If we change the dynamics to the usual time scale, each SGD step correspond to a time period $\eta$, an additional $\eta$ will appear on the numerator of the dynamics. Therefore, the dynamics is still a slow dynamics that is an $\eta$ factor slower than the original SGD dynamics.

**Case $\theta = \frac{\pi}{4}$.** When $\theta = \frac{\pi}{4}$, by equation 43, the dynamics of $u$ and $v$ are

$$du_t = c_2 \left( \frac{h(x'_t)}{2} - u_t \right) dt,$$

$$dv_t = c_2 \left( \frac{h(x'_t)}{2} - v_t \right) dt.$$

In this case, $u_t$ and $v_t$ have the same dynamics. If we assume $u_0 = v_0$, then we have $u_t = v_t$ for any $t \geq 0$. Then,

$$P_t^{-1} = \begin{bmatrix} \frac{1}{\sigma_0 \sqrt{u_t} + \epsilon_0 \sqrt{\gamma_2(t)}} & 0 \\ 0 & \frac{1}{\sigma_0 \sqrt{v_t} + \epsilon_0 \sqrt{\gamma_2(t)}} \end{bmatrix} = \frac{1}{\sigma_0 \sqrt{u_t} + \epsilon_0 \sqrt{\gamma_2(t)}} I.$$

Hence,

$$R_{-\theta} P_t^{-1} R_\theta = \frac{1}{\sigma_0 \sqrt{u_t} + \epsilon_0 \sqrt{\gamma_2(t)}} R_{-\theta} R_\theta = \frac{1}{\sigma_0 \sqrt{u_t} + \epsilon_0 \sqrt{\gamma_2(t)}} I.$$

Therefore, equation 43 can be written as

$$dx'_t = -\frac{\sqrt{\gamma_2(t)}}{\gamma_1(t)} \frac{m_{x',t}}{\sigma_0\sqrt{u_t} + \epsilon_0\sqrt{\gamma_2(t)}} dt$$

$$dy'_t = -\frac{\sqrt{\gamma_2(t)}}{\gamma_1(t)} \frac{m_{y',t}}{\sigma_0\sqrt{u_t} + \epsilon_0\sqrt{\gamma_2(t)}} dt$$

$$dm_{x',t} = c_1(h'(x'_t)y'^2_t - m_{x',t})dt$$

$$dm_{y',t} = c_1(2h(x'_t)y'_t - m_{y',t})dt + \sigma_0 c_1\sqrt{h(x'_t)}dB_t$$

$$du_t = c_2\left(\frac{h(x'_t)}{2} - u_t\right)dt. \tag{51}$$

The quasistatic step here still deals with the system of $y'$ and $m_{y'}$, and the results take the same form. we have

$$\lim_{\tau\to\infty} \mathbb{E}y'^2_\tau = \frac{\sigma_0^2\sqrt{\gamma_2(t)}}{4\gamma_1(t)(\sigma_0\sqrt{u_t} + \epsilon_0\sqrt{\gamma_2(t)})},$$

and the following effective dynamics on the minima manifold:

$$dx'_t = -\frac{\sqrt{\gamma_2(t)}}{\gamma_1(t)} \frac{m_{x',t}}{\sigma_0\sqrt{u_t} + \epsilon_0\sqrt{\gamma_2(t)}} dt$$

$$dm_{x',t} = c_1\left(\frac{\sigma_0^2\sqrt{\gamma_2(t)}h'(x'_t)}{4\gamma_1(t)(\sigma_0\sqrt{u_t} + \epsilon_0\sqrt{\gamma_2(t)})} - m_{x',t}\right)dt$$

$$du_t = c_2\left(\frac{h(x'_t)}{2} - u_t\right)dt. \tag{52}$$

To simplify the manifold dynamics above, we take the similar approximation steps as did for the $\theta = 0$ case. We first solve the $u$ dynamics, which gives

$$u_t = u_0 e^{-c_2 t} + \frac{c_2}{2}\int_0^t e^{c_2(s-t)}h(x'_s)ds.$$

Still assume $t$ is large, and the change of $h(x'_s)$ is slow compared with $e^{-t}$. Then, we can approximately take

$$\gamma_1(t) = 1, \quad \gamma_2(t) = 1, \quad u_t = \frac{h(x'_s)}{2}.$$

Substituting the above approximations into equation 52, we obtain

$$dx'_t = -\frac{m_{x',t}}{\sigma_0\sqrt{\frac{h(x'_t)}{2}} + \epsilon_0}$$

$$dm_{x',t} = c_1\left(\frac{\sigma_0^2 h'(x'_t)}{2\sqrt{2}\sigma_0\sqrt{h(x'_t)} + 4\epsilon_0} - m_{x',t}\right)dt. \tag{53}$$

Dropping the $\epsilon_0$ terms on the denominator, equation 53 is approximated by

$$dx'_t = -\frac{\sqrt{2}m_{x',t}}{\sigma_0\sqrt{h(x'_t)}}$$

$$dm_{x',t} = c_1\left(\frac{\sigma_0 h'(x'_t)}{2\sqrt{2h(x'_t)}} - m_{x',t}\right)dt. \tag{54}$$

Finally, if we assume $m_{x'}$ is close to its stationary solution, i.e. $m_{x',t} = \frac{\sigma_0 h'(x'_t)}{2\sqrt{2h(x'_t)}}$, we have the following first-order dynamics for $x'$ that approximates the manifold dynamics:

$$\frac{dx'_t}{dt} = -\frac{h'(x'_t)}{2h(x'_t)}. \tag{55}$$

The dynamics equation 55 is a gradient flow that minimizes $\ln h(x)$ on the minima manifold. Again, due to the time scale choice, the dynamics gets slower for smaller learning rate. Compared with the dynamics equation 50 for the case $\theta = 0$, this dynamics is slower because there is no $\epsilon_0$ on the denominator.

**Remark 4.** *The approximation steps in the derivations above are conducted intuitively without rigorous proof. The goal is to unveil and compare the essential components of the dynamics. If strict theorems are to be proved, conditions and assumptions need to be imposed.*

