# OpenReview forum: "A Quasistatic Derivation of Optimization Algorithms' Exploration on Minima Manifolds"
_ICLR.cc/2023/Conference — Submitted to ICLR 2023_

### Official Review · Reviewer_KFCz · 2022-10-16

**Confidence:** 3
**Correctness:** 1
**Technical Novelty And Significance:** 2
**Empirical Novelty And Significance:** 2
**Recommendation:** 3

**Clarity, Quality, Novelty And Reproducibility:**

Some arguments are misleading, the analysis is vague and not supported by solid theory and empirical studies.

**Strength And Weaknesses:**

1. There are a number of assumption used without being justified. For example, it is assumed that the Hessian has zero eigenvalues along the tangent space of $\mathcal{M}$. And in eq(18), it is assumed that the momentum is always at equilibrium. How to justify these assumptions for over-parameterized deep neural network?

2. The claims are not well supported by theory, instead it is more of some intuition for some specific problems, not for over-parameterized neural network as claimed. For example, the loss function is approximated by a squared function based on Hessian matrix. But how accurate is this approximation for a deep neural network? In what range is this approximation accurate? These are all critical components of the analysis and they need rigorous analysis.

3. The study in Section 3 and Section 5 are for quadratic convex problems which cannot directly provide intuition for deep neural networks which have much more complicated landscape.

4. The experiments are not adequate at all to verify the proposed approach. Especially considering a rigorous theoretical analysis is lacked, it takes solid empirical studies to make the results convincing.





**Summary Of The Paper:**

This submission proposes a quasistatic approach to derive the optimization algorithms's behavior on the manifold of minima. It has tried to understand the role of some parameters such as learning rate and batch size. Some unrealistic assumptions have been made, the results are not supported by rigorous analysis and are not well verified by solid empirical studies.

**Summary Of The Review:**

Some unrealistic assumptions have been made, the results are not supported by rigorous analysis and are not well verified by solid empirical studies.

---

### Official Review · Reviewer_fp5w · 2022-10-21

**Confidence:** 3
**Correctness:** 3
**Technical Novelty And Significance:** 2
**Empirical Novelty And Significance:** 2
**Recommendation:** 5

**Clarity, Quality, Novelty And Reproducibility:**

- I’m confused about the contribution this paper makes in addition to Li et al. The main point is a new method to derive their results? Even the core idea of separation of timescales in the SDE are mentioned in Li et al. Are there new machine learning implications that are found using this analysis?

- Are s and r novel to this paper?

- The clarity of the paper would be improved by a more organized statement about which results require which assumptions.


**Strength And Weaknesses:**

**Strengths**

- The paper is clearly written with nice illustrative, simple examples to explain the core ideas.
- Fig 3 is nice. How are the s and r kept the same in (Middle right)? Which hyperparameters are changed between the two models?


**Weakness**

- The assumptions seem to be doing a lot of work in the analysis. For example, alignment of the noise covariance and the Hessian is frequently assumed. It's not clear to me how reasonable such assumptions are for real models.
- On page 4: "In many cases, the noise covariance matrix of SGD aligns with the Hessian Mori et al. (2021)." Is this an empirical fact?
- My main concern is novelty.

**Summary Of The Paper:**

The authors use a quasistatic approach to analyze an SDE related to SGD near minima of the loss landscape. They find a coupled SDE that has two component, one tangent and another normal to the manifold of minima. The drift terms in the equations lead to a separation of timescales where the normal component is significantly faster and is treated as being in equilibrium. Plugging in the equilibrium behavior of the normal component into the SDE for the tangent component's equation reveals an implicit regularization term that is minimized by SGD. The authors apply this analysis to SDEs modeling SGD, SGD with momentum, and Adam.

**Summary Of The Review:**

I enjoyed reading the paper and feel it contains some nice results. I don't feel all the implications of the analysis are tied back to realistic models---more could be done in this direction. The main concern is novelty, especially with respect to Li et al. I would be happy for the other reviewers to comment on this.

---

### Official Review · Reviewer_pGAs · 2022-10-23

**Confidence:** 3
**Correctness:** 4
**Technical Novelty And Significance:** 3
**Empirical Novelty And Significance:** 3
**Recommendation:** 5

**Clarity, Quality, Novelty And Reproducibility:**

Technically the paper seems to be solid, but I have not checked the math in detail. As regards the clarity, I think that the writing of the paper can be improved such that to make it more accessible to the reader (see questions), while in my opinion the writing of the current version is a bit hasty.

**Strength And Weaknesses:**

The problem of understanding the learning dynamics of stochastic optimizers, especially for over-parametrized neural networks, that the paper aims to address is indeed important in the field. The authors introduce this problem mainly based on works related to deep networks, and even if I am not an expert in this field, I think that the related work is decently presented. I find the motivation for the proposed approach ok, but I believe that perhaps this can be improved (see questions). The high level idea of the approach is interesting and seems sensible, but in some parts the report is rather confusing and I think hard to follow in details (see questions).

**Summary Of The Paper:**

The authors provide a technique to approximate the dynamics of stochastic optimizers that typically oscillate around a manifold, which corresponds to the minima of the loss. This allows to study and compare the properties of stochastic optimizers, for example the tendency to search for flatter minima. In the experiments empirically compare the behavior of the proposed approximation to the true dynamics, and demonstrate the ability to study properties of the optimizer.

**Summary Of The Review:**

Questions:

1. I think that some illustrations can help significantly and make the paper much more accessible, as there are a lot of technical details that can be probably summarized accordingly.

2. What is the definition of the minima manifold? Is it the set in the parameter space where the loss function is zero? What is the effective/manifold dynamics? Perhaps these terms should be defined explicitly.

3. I believe that a description of the quasistatic dynamics should be included at least in the appendix, while now the context is roughly presented around Eq. 2.

4. For Eq. 2 the quasistatic means that we have a coupled system of differential equations, and near the minima manifold the $y_t$ moves faster than $x_t$, where "faster" simply means that the step along the y-axis is bigger? The implied trajectory of the derived system is for every time step $t$ a point $y_t$ sampled from the associated Gaussian while the $x_t$ follows the dynamics of the equation after Eq. 2?

5. In general, what is the intuition of the dynamical system after applying the quasistatic derivation? Can we still dicretize the process and get a discrete trajectory?

6. Is the high level idea of your approach that according to your approximation, when we are near the minima manifold, the optimizer moves along this manifold searching for a better minimum, while along the normal direction the optimizer moves "faster" (jumping along the normal direction), but in expectation we remain near the manifold?

7. The derived dynamics are not for the actual loss function, but for the surrogate function $f$ (Eq. 5) that is a quadratic approximation to the true loss (using Taylor's expansion)?

8. I suppose that the particular structure of the Hessian with many zeros, holds only when the minimizer is on an axis of $\mathbb{R}^d$ e.g. [x, 0]. Because, even if we have a flat minima manifold that is not axis aligned, then the particular Hessian does not necessarily have this particular structure. How the analysis changes if the Hessian does not have all these zeros? Probably some extra linear terms $y_\tau$ appear and their expectation goes to 0?

9. How the analyis changes if the the assumption about the uncorrelated noise (between tangent and normal space) does not hold?

10. I think that it is not clear where the discussion for the orthonormal tangent spaces at the end of page 4 is actually used.

11. For the general manifold it is assumed that the point $z_0$ is axis aligned, so the Hessian has the structure with zeros. How the analysis changes if this is not the case?

12. What is the intuition for the companion loss function $\tilde{f}$ ? If I understood correctly this function takes points around the $T_{z_0}\mathcal{M}$, maps them around the $\mathcal{M}$, and then these points are projected to $\mathcal{M}$ such that to use the corresponding Hessian of the final point?

13. The actual implication for Eq. 10 and Eq. 11 is that $z_0$ lies on the axis and in a local neighborhood the minima manifold can be seen as flat? So the actual analysis for the general manifold concludes that in a local neighborhood the dynamics are exactly the same as the flat manifold, as the manifold locally is considered flat?

14. Does the $\nabla_\mathcal{M}$ implies that we actually compute the gradient on the manifold which returns a "tangent vector"? Also, what is the dimensionality of Eq. 12?

15. Could you provide some empirical evidences for deep networks?

16. In the "Example" (Eq. 13) it is implied that the Hessian has this particular structure with zeros, and for this reason the $\tilde{D}_{11} = 0$, but I think that this only holds if the minima are axis aligned?

In general, I like theoretical works that analyze the problem in an accessible manner. I think that the proposed idea has some very interesting aspects. However, in my opinion the current version of the paper is quite confusing in some parts. I acknowledge the fact that theoretical work is typically harder to explain and simplify, but I believe that some improvements can be made. In my opinion, the current version is not ready for publication mainly for clarity and accessibility issues.

---

### Decision · Program_Chairs · 2023-01-20

**Decision:**

Reject

**Justification For Why Not Higher Score:**

NA

**Justification For Why Not Lower Score:**

NA

**Metareview: Summary, Strengths And Weaknesses:**

The papers proposes a quasistatic approach to study an SDE related to the behavior of stochastic gradient descent (SGD) near minima in the loss landscape of a neural network. They find that the SDE can be split into two components: one that is tangent to the manifold of minima and one that is normal to it. They show that the normal component of the SDE reaches equilibrium more quickly than the tangent component, and use this separation of timescales to derive an implicit regularization term that is minimized by SGD. They also apply this analysis to SGD with momentum and Adam, two other optimization algorithms commonly used in machine learning.

The reviewers raised several concerns, but no responses were given by the authors.